# Coupling of a Novel TIMP3 Peptide to Carboxypeptidase G2 for Pro-Drug Activation at the Tumour Site

**DOI:** 10.3390/molecules26030625

**Published:** 2021-01-25

**Authors:** Mohammed S. Aldughaim, Fatimah Alsaffar, Michael D. Barker

**Affiliations:** 1Research Centre, King Fahad Medical City, P.O. Box 59046, Riyadh 11525, Saudi Arabia; 2Department of Oncology and Metabolism, Medical School, University of Sheffield, Beech Hill Rd, Sheffield S10 2RX, UK; falsaffar@gc.edu.sa (F.A.); m.barker@sheffield.ac.uk (M.D.B.); 3Department of Clinical Laboratory Sciences, Alghad International Colleges for Applied Medical Sciences, Dammam 32423, Saudi Arabia

**Keywords:** angiogenesis, carboxypeptidase G2, TIMP3, VEGFR2

## Abstract

Broad-spectrum cytotoxic drugs have been used in cancer therapy for decades. However, their lack of specificity to cancer cells often results in serious side-effects, limiting efficacy. For this reason, antibodies have been used to attempt to specifically target cytotoxic drugs to tumours. One such approach is antibody-directed enzyme prodrug therapy (ADEPT) which uses a tumour-directed monoclonal antibody, coupled to an enzyme, to convert a systemically administered non-toxic prodrug into a toxic one only at the tumour site. Among the main drawbacks of ADEPT is the immunogenicity of the antibody-enzyme complex, which is exacerbated by slow clearance due to size, hence limiting repeated administration. Additionally, the mono-specificity of the antibody could potentially result in drug resistance with repeated administration. We have identified a novel short peptide sequence, p700, derived from a human tissue inhibitor of metalloproteinases-3 (TIMP-3), which binds to and inhibits a number of tyrosine kinase growth factor receptors (VEGFRs1-3, FGFRs 1-4 and PDGFRα) which are known to be upregulated in many tumours and tumour vasculature. In this report, we fused p700 to His-tagged, codon-optimised, carboxypeptidase G2 (CPG2). CPG2 is a bacterial enzyme used in ADEPT, which activates potent nitrogen-mustard pro-drugs by removal of an inhibitory glutamic acid residue. Recombinant CPG2-p700 was highly expressed in *Escherichia coli* and successfully purified by nickel affinity chromatography. Biolayer interferometry showed that CPG2-p700 had a 100-fold increase in binding affinity for VEGFR2 compared with CPG2 alone and retained its catalytic activity, as determined by methotrexate cleavage. In the presence of CPG2-p700, the ZD2676P pro-drug showed significant cytotoxicity for 4T1 cells compared with prodrug alone or CPG2 alone. p700 is, therefore, a potentially useful alternative to monoclonal antibodies for enzyme pro-drug therapy and could equally be used for effective delivery of other cytotoxic drugs to tumour tissue.

## 1. Introduction

Broad-spectrum cytotoxic drugs, such as nitrogen mustards, have been the mainstay of cancer therapy for many years. However, such drugs not only target cancer cells but all proliferating cells, resulting in severe side-effects, which limits dosage and potential efficacy in the long-term [1]. Many mechanisms have been used to target chemotherapeutics specifically to the cancer sites, including antibody directed enzyme prodrug therapy (ADEPT) [2,3,4]. In ADEPT, enzymes that convert prodrugs to active drugs are first targeted to the cancer site by a tumour-specific monoclonal antibody. After clearance of the enzyme from normal tissue, the prodrug is administered to be activated at the cancer site [5]. ADEPT is advantageous due to the accumulation of the enzyme–antibody conjugate within the tumour vasculature after clearance from normal tissues.

A *Pseudomonas aeruginosa* strain RS-16-derived enzyme, carboxypeptidase G2 (CPG2 or glucarpidase), has been used in this staged therapy [6]. This is partly because its activity is not found in humans, reducing the chance of toxicity to healthy tissue, as the prodrug will be activated only by the localized exogenous enzyme. This zinc-dependent enzyme naturally catalyses the hydrolysis of the C-terminal glutamic acid residue of folic acid and synthetic folate analogues such as the cancer chemotherapy agent, methotrexate (MTX). For this reason, CPG2 is sometimes used clinically for clearing excess MTX in patient blood after high dose therapy to control its side effects [7,8]. In ADEPT, CPG2 can be used to activate prodrugs such as nitrogen mustard l-glutamate prodrugs into nitrogen mustards, which cross-link DNA leading to apoptosis [9].

However, there are several drawbacks to ADEPT that have prevented its successful application in the clinic. Most importantly, the host immune response to this conjugate is a major problem in the successful application of ADEPT preventing repeat dosage and leading to neutralisation of the antibody [3,4]. This is exacerbated by slow clearance rates of the complex. Prolonged circulation times are also a major issue, as the enzyme must be cleared from the circulation prior to pro-drug administration to avoid off target activation. While antibodies can be humanised, this is an expensive and time-consuming process. Additionally, antibodies are mono-specific, and tumours often rapidly evolve to lose expression of the target molecule. While several different antibodies could be used, this increases the complexity and cost, particularly if humanisation is required [10].

As an alternative to using monoclonal antibodies, in this study, we utilised a short, tumour-specific peptide, p700, to target the enzyme CPG2 to tumours. p700 is a short 16 amino acid fragment derived from the C terminal domain of tissue inhibitor of metalloproteinase-3 TIMP3, which potently inhibits VEGFR2. Unlike the parent molecule, however, p700 exhibits a broader binding specificity for other growth factor receptors such as VEGFR1, VEGFR3, PDGFRα, FGFR1, FDFR2α, FDFR3 and FDFR4, all of which are highly expressed in tumours or tumour vasculature [11]. Using this peptide instead of a monoclonal antibody has several potential advantages. Firstly, the peptide is derived from a human extracellular protein and is not expected to be immunogenic, even in mice which share the same sequence at this site. Secondly, the small size of the peptide should enable much more rapid clearance of the complex, further reducing any potential immunogenicity and non-targeted prodrug activation. Lastly, the peptide targets multiple receptors on both tumours and tumour vasculature, greatly decreasing the likely development of drug resistance.

## 2. Methods

### 2.1. Synthesis of CPG2 Codon-Optimized Synthetic Gene

A CPG2 gene bearing a 3′[His]_5_ tag and codon optimised for expression in *E. coli* (Goda et al. 2009) was synthesised by Eurofins Genomics (UK). However, the second *AgeI* site at base 1050 was engineered out to leave a single *AgeI* site at position 106, while 5′ *NdeI* and 3′ *HindIII* and *NotI* restriction sites were added to facilitate cloning into pET28a bacterial expression vector. This was supplied in a kanamycin resistant cloning vector; pEX-K4. The p700 peptide sequence with 5′ *NdeI* sites was also synthesised by Eurofins, including a [Gly_4_Ser]_4_ linker and the CPG2 sequence up to the *AgeI* site and supplied in an ampicillin-resistant cloning vector, pEX-A2. All the plasmids were transformed into α-Select chemically-competent *E. coli*, followed by mini-prep purification (Qiagen Manchester, UK) following the manufacturer’s instructions.

### 2.2. Sub-Cloning of the Codon Optimised CPG2-His Gene into pET28a and Insertion of the p700 Peptide Sequences

Codon-optimised CPG2-His (approximately 1200 bp in length) was excised from pEX-K4 by digestion with *NdeI* and *NotI* and ligated into the bacterial expression vector, pET28a, to form an unmodified CPG2-His-pET28a expression construct. The p700-[Gly_4_Ser]_4_ sequence was then excised from the pEX-A2 vector and cloned into the *NdeI* and *AgeI* restriction sites of the CPG2-His-pET28a by cutting both constructs with these enzymes to generate a p700-[Gly_4_Ser]_4_-CPG2-His-pET28a expression construct (refer to Appendix A). Both constructs were transformed into α-Select chemically competent *E. coli* (Bioline, London, UK). Miniprep and maxiprep plasmid purifications (Qiagen, Manchester, UK) were then performed.

### 2.3. Small Scale Induction of CPG2-Only and CPG2 Fusion Protein Expression

Following verification of the two CPG2 constructs’ DNA sequences, plasmid DNA was transformed into *E. coli* BL21 (DE3) grown on LB agar plates containing 50 µg/mL kanamycin and 0.2% glucose. A single colony was then transferred to a tube containing 10 mL LB broth and 50 µg/mL kanamycin and incubated overnight at 37 °C on a rotating shaker.

For induction, 9 mL of these cultures was then inoculated into 100 mL of LB broth (plus antibiotic) and further incubated until the optical density (OD) reached 0.5–0.6. Isopropyl-β-d-thiogalactopyranoside (IPTG) was then added at a final concentration of 1 mM to induce protein expression. This was followed by further incubation, and samples (1 mL) were then removed at 1, 2, 3 and 4 h post-induction.

### 2.4. Protein Detection by SDS-PAGE and Coomassie Brilliant Blue Staining

Pellets obtained by centrifuging the above cultures at 14 × 10^3^× *g* were re-suspended in 50 µL dH_2_O, 10 µL of which was diluted with an equivalent volume of 2 × Laemmli sample buffer (Bio-Rad^®^ Kidlington, UK) and 1 µL of 1 M dithiothreitol (DTT). Sample tubes were heated at 95 °C for 5 min, after which the samples were transferred to QIAshredder tubes and centrifuged at 14 × 10^3^× *g* for 2 min to remove debris and shear the genomic DNA.

An amount of 15 µL of the above samples was separated by SDS-PAGE on a 10% gel, followed by staining with colloidal Coomassie G250 stain (National Diagnostics) and then imaging using a Bio-Rad^®^ Gel DOC^TM^ EZ imager.

### 2.5. Solubility Test of Recombinant Proteins

A solubility test was carried out to determine whether the expressed proteins were in the soluble fraction (supernatant) or the insoluble fraction (pellet) of the bacterial cell lysates. Briefly, IPTG-induced *E. coli*-expressing CPG2 and CPG2-fusion proteins were grown in LB broth supplemented as above for 4 h, after which a 1 mL sample was taken and centrifuged at 14 × 10^3^× *g* for 2 min. The pellet was resuspended in 2 mL of dH_2_O containing Roche Protease Inhibitor Cocktail (1 tablet in 50 mL), followed by 40 µL of lysozyme (10 mg/mL), mixed and then incubating on ice for 20 min. Next, 80 µL of sodium deoxycholate (25 mg/mL) was added followed by incubation on ice as above. The cells were lysed by sonication (4 cycles at 10 s/cycle) on ice. The lysate was centrifuged at 14 × 10^3^× *g* for 25 min (at 4 °C), and the collected pellet was resuspended in 2 mL dH_2_O. Whole cell lysate, supernatant and resuspended pellet were analysed on a Coomassie Blue stained SDS-PAGE gel.

### 2.6. Large Scale Induction of Recombinant CPG2 Expression

Large scale cultures of transformed BL21 cells were prepared as before, but each starter culture was diluted into 1 L of LB broth containing 50 µg/mL kanamycin and grown at 37 °C with shaking. At an OD of 0.5–0.6, the cells were induced with IPTG to a 1 mM final concentration with a 1 mL sample taken before induction. Cells were incubated for a further 3–4 h at 37 °C with shaking. The cells were harvested at 5500× *g* for 15 min at 4 °C. The pellets were processed as described below.

### 2.7. Whole Bacterial Cell Lysis and Separation into Soluble and Insoluble Fractions

Pellets from the large-scale expression (1 L culture) were resuspended in 20 mL Tris buffer (20 mM Tris, 137 mM NaCl, 1 mM EDTA pH 7.6) containing protease inhibitor cocktail (Roche, Welwyn Garden City, UK). Cells were lysed with 400 µL of 10 mg/mL lysozyme (final concentration of 200 µg/mL) and incubated on ice for 30 min. An amount of 800 µL of 25 mg/mL sodium deoxycholate detergent was added to the lysate and incubated on ice for 20 min, followed by four 1 min sonication cycles. Cell lysates were then centrifuged at 15,000× *g* for 25 min at 4 °C and the supernatants and pellets retained.

### 2.8. Inclusion Body Preparation

#### 2.8.1. Insoluble Fraction

The above pellets were resuspended in 20 mL of Tris buffer with urea (20 mM Tris, 137 mM NaCl, 2 M urea, 1 mM EDTA pH 8) and mixed thoroughly. Aliquots of 50 µL of the suspension were retained for SDS-PAGE analysis.

#### 2.8.2. Washes

The remaining cell suspensions were then centrifuged at 15,000× *g* for 30 min. The supernatants (Wash 1) were collected and the pellets resuspended in Tris buffer with urea for three other rounds of centrifugation. The second supernatants were also collected for SDS-PAGE analysis.

#### 2.8.3. Inclusion Body

The final pellet (inclusion body) was transferred into a microfuge tube and re-suspended in 50 µL of dH_2_O for SDS-PAGE analysis.

### 2.9. CPG2 Purification from the Insoluble Fraction by Ni^2+^-NTA Chromatography

#### 2.9.1. Preparation of Ni^2+^ Column

The Ni^2+^-NTA columns (Sigma-Aldrich, Gillingham, UK) were initially equilibrated with 20 mL dH_2_O running under gravity followed by charging with 10 mL of 10 mM NiCl_2_, and then 20 mL of water running under gravity. The column was then washed with 10 mL of wash buffer 1 (20 mM Tris, 137 mM NaCl, 2 M urea; pH 8).

#### 2.9.2. Protein Purification

Washes 1–4 were pooled together (making a total of approximately 60 mL) and 300 µL of 4 M imidazole added to a final concentration of 20 mM and loaded onto the Ni^2+^-NTA columns. Columns were then washed with 10 mL of filtered wash buffer 2 (20 mM Tris, 137 mM NaCl, 1 M urea; pH 8) and the flow through collected each time for SDS-PAGE analysis to ensure that the protein had bound to the columns.

#### 2.9.3. Elution

His tagged proteins bound to the columns were then eluted with 10 mL elution buffer 1 (20 mM tris, 137 mM NaCl, 500 mM urea, 200 mM imidazole; pH 8). The eluents were collected in 1 mL fractions for SDS-PAGE analysis. Further elution was performed by addition of 10 mL elution buffer 2 (20 mM Tris, 137 mM NaCl, 500 mM urea, 500 mM imidazole; pH 8), and samples collected as above.

### 2.10. Enzyme Activity Assay

Recombinant CPG2 fusion protein activity was assessed using methotrexate (MTX) as substrate with a modified protocol of Goda, Rashidi [5]. Briefly, 1 mL of 100 mM Tris-HCl pH 7.3 containing 0.2 mM ZnSO_4_ and 60 µM of MTX was added to a cuvette and incubated at 37 °C for 5–10 min. Purified recombinant CPG2 fusion protein was then added (final concentration of 16 µg/mL) to the cuvettes, which were placed directly into a spectrophotometer, maintained at 37 °C. Tris buffer without MTX was used as background control, and decrease in absorbance at 320 nm was measured at 5 min intervals.

### 2.11. Assessment of Zinc-Dependence of Recombinant CPG2-Fusion Protein Enzyme Activity

Zinc-dependence of the purified recombinant CPG2 fusion proteins for enzyme activity was evaluated as described above in the presence of ZnSO_4_ or in its absence with the addition of 10 mM EDTA as a zinc chelating agent.

### 2.12. Kinetic Analysis of CPG2-p700 Binding to VEGFR2 by Biolayer Interferometry (BLI)

Biolayer interferometry (BLI), using the BLItz^®^ system (ForteBio, Göttingen, Germany), was used for real-time kinetic analysis of the interaction between recombinant CPG-p700 and its ligand, VEGFR2, according to the manufacturer’s instructions.

Briefly, protein A-coated probes were hydrated by soaking in PBS for at least 10 min and then loaded onto the biosensor mount. An initial baseline was established by lowering the probe into 250 µL of PBS in the tube holder for 60 s. VEGFR2-Fc ligand solution (R&D Systems), 5 µL at a concentration of 250 μg/mL, was then loaded into the drop holder and the biosensor inserted for 300 s. A further baseline step was then established before the CPG2 recombinant proteins (CPG2 or CPG2-p700) were introduced into the drop holder for the association and dissociation steps. Control runs were performed with only PBS and only VEGFR2-Fc (no recombinant CPG2 proteins added) for comparison.

The experiment was then repeated five times with subsequent sample (CPG2 or CPG2-p700) concentrations ranging from 600 nM to 10 µM. The binding affinity (Kd) was then calculated by the BLItz^®^ software based on the binding signal (nm), sample concentration and molecular weight.

### 2.13. Cytotoxicity Assay

The ability of the purified CPG2 recombinant proteins to activate the nitrogen mustard pro-drug ZD2676P ((2*S*)-2-[(4-[bis(2-iodoethyl)amino]phenoxy-carbonyl)amino] pentanedioic acid hydroiodide—High Force Research, Durham, UK) was evaluated using a cytotoxicity assay with the mouse breast cancer cell line 4T1.

4T1 cells were seeded into three 96-well plates at a density of 2 × 10^3^ cells per well and then cultured overnight at 37 °C. Three 96-well plates were prepared, one with 100 µL medium containing recombinant codon-optimised CPG2-only protein at 16 µg/mL maximal enzyme activity concentration, a second containing recombinant CPG2-p700 at the same concentration (16 µg/mL) and a third with 100 µL fresh medium only. The plates were incubated for 4 h at 37 °C to allow binding of the CPG2-p700. The media in all wells were then aspirated completely and the cells washed twice with PBS to remove any unbound CPG2 residues.

Triplicate wells of the cells were exposed to varying concentrations of ZD2676P prodrug ranging from 2 to 14 µM concentrations and incubated at 37 °C for one hour (as recommended by the pro-drug supplier). The media were aspirated, replaced with fresh growth medium and the plates incubated for 3–4 days before 20 µL of MTS solution (CellTiter 96^®^ AQ_ueous_ One Solution-Promega, Southampton, UK) was added into each well to determine cell viability, according to the manufacturer’s protocol. The absorbance of the wells was recorded at 490 nm with a 96-well plate reader. For each cell line, experiments were repeated three times in six wells of each 96-well plate. Statistical analysis was performed using two-way ANOVA and Bonferroni multi-comparison. Percentage cell viability curves were drawn using GraphPad prism software (v7.0).

## 3. Results

### 3.1. Solubility Test

Solubility testing was carried out to determine whether the CPG2 was expressed predominantly in the cytosol (soluble fraction) or in inclusion bodies (insoluble fraction). After 3 h of IPTG induction, expression of the CPG2 and CPG2-p700 was found to be minimal in both initial whole cell lysate and in the soluble fraction of the cells which was derived from supernatants of centrifuged whole cell lysate. However, the insoluble fraction derived from pellets of centrifuged whole cell lysate showed high expression of the CPG2 and CPG2-p700 proteins (Figure 1).

### 3.2. Large Scale Expression of CPG2 and CPG2 Fusion Proteins

After determining that the recombinant proteins were maximally expressed after 3 h, a large-scale IPTG-induced expression of the recombinant proteins was carried out to determine if the expression could be replicated in bulk. As shown by both Coomassie blue staining (Figure 2A) and Western blotting (Figure 2B), there is clear, high-level expression of CPG2 and CPG2-p700 after 3 h.

### 3.3. Inclusion Body Preparation for CPG2 and Purification of CPG2 Recombinant Proteins

The solubility testing confirmed that the recombinant proteins localised almost exclusively to the inclusion bodies of the bacteria which would necessitate extraction of the proteins with a denaturing agent that would permit refolding of functional proteins. Urea is commonly used for this purpose and, in the protocol for CPG2 expression described by Goda, Rashidi [5], the pellets were initially washed in three changes of 2 M urea before complete dissolution in 6 M urea, and so this procedure was replicated here. However, in our case, considerable amounts of relatively pure recombinant proteins were found in the 2 M urea washes, as shown in Figure 3A,B. While most of the protein was still found in the remaining pellets, it was thought that the protein found in the washes may not be fully denatured and would probably be sufficient to carry out initial functional screening assays, while the protein in the remaining pellets would require much more aggressive denaturation, and subsequent renaturation, which may result in loss of function.

As alluded to above, instead of extracting the CPG2 recombinant proteins from the pellets remaining after the urea washes, the recombinant proteins were purified from the four pooled 2 M urea washes using Ni-NTA affinity chromatography. The result of the purification step is shown in Figure 3C,D, and we were able to acquire sufficient purified protein (1 mg/mL) for subsequent functional assays.

### 3.4. Assessment of the Enzymatic Activity of the Purified Recombinant CPG2proteins

The catalytic activity of the purified proteins was assessed using a methotrexate (MTX) cleavage assay. As CPG2 is a zinc-dependent enzyme, the assay was carried out in the presence or absence of zinc, and in the presence of zinc and EDTA (a zinc chelator) in order to confirm that any cleavage seen was most likely due to CPG2 and not any other potentially contaminating protein. MTX was used as it is readily cleaved by CPG2 resulting in a reduction in its absorbance at a wavelength of 320 nm which can be observed spectrophotometrically. Figure 4 shows that both CPG2 and CPG2-p700 were able to metabolise MTX in the presence of zinc, but not in its absence nor in the presence of zinc and EDTA. The rate of breakdown was almost identical between CPG2 and CPG2-p700, indicating that the purified enzyme is both functionally active and that the p700 sequence does not interfere with this activity.

### 3.5. Determination of the Ability of CPG2-p700 Recombinant Protein to Bind to VEGFR2 by Biolayer Interferometry

Biolayer interferometry enables kinetic analysis to be carried out on very small sample volumes and was used to confirm that covalent coupling of p700 to CPG2 did not interfere with its ability to bind to one of its key target receptors, VEGFR2. A VEGFR2-immunoglobulin Fc fusion protein was used as it enabled ready binding of VEGFR2 to the protein A sensor chips of the BLItz machine in the correct orientation. Analysis of the association and dissociation data at a range of CPG2 concentrations indicated that p700 increased the affinity of CPG2 for VEGFR2 by about 100-fold, with a calculated dissociation constant (Kd) of 130 nM compared to 15 μM for CPG2 alone. A representative experiment, at a single protein concentration, is shown in Figure 5. This confirmed that p700 facilitates high affinity binding of CPG2 to VEGFR2.

### 3.6. Cytotoxicity Assays to Determine the Ability of Recombinant CPG2 Proteins to Activate the ZD2676P Prodrug

The effectiveness of both CPG2 and CPG2-p700 to convert the nitrogen mustard prodrug, ZD2676P, into a cytotoxic drug, and thus kill 4T1 mouse breast tumour cells, was evaluated using an MTS cytotoxicity assay. Using a range of concentrations of the prodrug, based on a previous study [12], it was found that there was no significant difference in cell death between cells treated with ZD2676P alone or those treated with ZD2676P and CPG2 enzyme. However, at the highest dose of ZD2676P (14 μM), there was a significant increase in cell death with cells treated with CPG2-p700, relative to both prodrug alone and prodrug with unlabelled CPG2 (Figure 6).

## 4. Discussion

Drug toxicity is a major challenge for anticancer chemotherapeutics and is the reason for the wealth of studies investigating delivery strategies that might suppress the toxic side effects of conventional drugs [13]. CPG2 is sometimes used in the clinic to reduce excess concentration of methotrexate in cancer patients by converting it into 4-deoxy-4-amino-N10-methylpteroic acid (DAMPA) and glutamate, which are both non-toxic and metabolised by the liver [14], and so is known to be safe in the clinic. CPG2 is also the only enzyme that has been tested in ADEPT clinical trials, which were not altogether successful [3].

We reasoned that the use of p700 to target CPG2 or other enzymes to tumour sites, rather than antibodies, could overcome some of the issues associated with these ADEPT trials, by potentially reducing immunogenicity, decreasing clearance times and decreasing potential drug resistance due to the ability of p700 to target multiple receptors [11]. Nevertheless, whether p700 would retain its ability to bind receptors once covalently coupled to an enzyme could only be determined empirically.

While there are several small molecule drugs that also target a wider family of tyrosine kinase receptors, such as sorafenib and sunitinib, these all target the intracellular kinase domain and can also target other intracellular kinases with the potential for off-target effects. Extracellular, competitive inhibitors of these receptors may decrease the likelihood of side-effects and also offer the potential to act as ligands for targeted drug delivery.

Previous data from our laboratory had shown that p700 could potently inhibit tumour growth in a syngeneic breast tumour model [11]. However, on its own, the molecule is not cytotoxic, and tumours were found to regrow following treatment cessation or prolonged treatment (unpublished observation). This fuelled the idea that, in addition to inhibiting target receptors, p700 might be used as a delivery vehicle to target cytotoxic drugs to tumour sites, potentially enhancing the therapeutic potential of p700 while decreasing the off-target effects of chemotherapy.

As such, we coupled p700 to the N-terminus of CPG2 via a [Gly_4_Ser]_4_ linker and showed that it could be purified from the insoluble fraction of bacteria in a form that retained both the catalytic activity of the CPG2 enzyme, as determined by methotrexate cleavage, and also the ability of p700 to bind to VEGFR2 with a high affinity, as determined by BLI. Additionally, p700 helped localise CPG2 to the surface of 4T1 tumour cells, increasing cytotoxicity in the presence of the ZD2676P prodrug.

Although we only tested binding of the CPG2-p700 complex to VEGFR2 (the target of the TIMP3 parent protein), and not the other closely related target receptors of p700, it would seem likely that this wider receptor binding profile is indeed retained, not only because these other receptors share a very similar ligand binding site in their D2/D3 immunoglobulin-like domains but also due to the fact that p700 appeared to effectively localise CPG2 to the surface of 4T1 cells, which do not express significant levels of VEGFR2 [15]; however, they do express VEGFR1 [16] and FGF receptors [17].

Although the improved ability CPG2-p700 to activate the pro-drug in the cytotoxicity assay, relative to CPG2 alone, was only seen at the highest dose of pro-drug, these concentrations were based on those from a previous study using a human colorectal cancer cell line and had not been optimised for 4T1 cells. A greater difference between the uncoupled- and p700-coupled CPG2 may have been seen if a higher dose range had been assessed. 4T1 is a highly metastatic breast cancer cell line that is very resistant to cytotoxic drugs [18], and so this lower level of sensitivity is not surprising. In this case, 4T1 cells were chosen as their growth in mice had previously been shown to be sensitive to the p700 peptide (although the peptide alone is not cytotoxic) [11], enabling future drug testing in vivo. Effective cell killing of these cells via p700-CPG2 is, therefore, a very robust test of its efficacy. Clearly, it will be important to reassess this efficacy on human tumour cells. However, the p700 region of TIMP3 does indeed bind to multiple pro-angiogenic growth factor receptors on human cells [11], and because it is totally conserved between mice and humans, it is also highly unlikely to be immunogenic in either species. Moreover, we have recently shown that p700 can effectively deliver liposomal doxorubicin to MCF7 human breast cancer cells [19]

While p700 is much less likely to elicit an immune response than a non-humanised monoclonal antibody, it is unlikely to reduce the potential immunogenicity of CPG2 itself. Nevertheless, our understanding of immunogenic epitopes is constantly improving, and it is possible to predictively “deimmunise” proteins, as has been shown for certain β-lactamases that have applications in ADEPT [20], so it is likely that immunogenicity of the enzyme may become less of an issue in the future.

## 5. Conclusions

The data presented here verify the feasibility of using p700 as an alternative to monoclonal antibodies in directing CPG2 to tumour cells, its key advantages being small size (<2 kDa versus 150 kDa), lack of immunogenicity and multi-receptor binding. Moreover, we have shown that the CPG2-p700 fusion protein can be produced as a single functional protein in a bacterial expression system, hugely decreasing the potential production costs of such a molecule over ADEPT, where enzyme and antibody must be produced separately and chemically cross-linked, and where any antibody found to be effective in animal studies would need to be humanised prior to trials in humans.

## Figures and Tables

**Figure 1 molecules-26-00625-f001:**
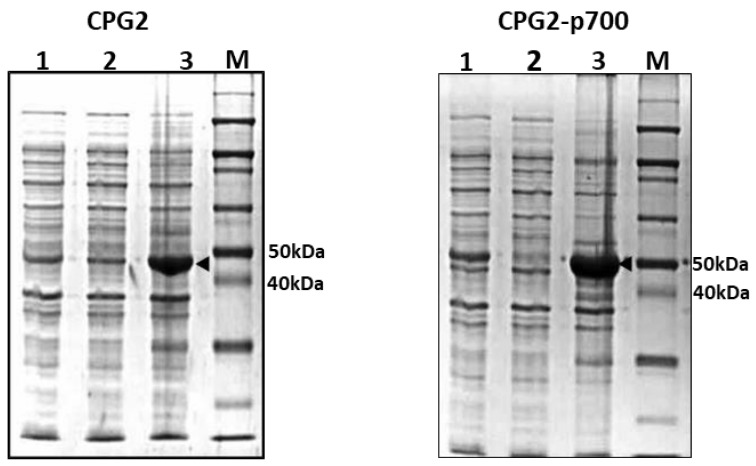
Solubility test to determine localisation of recombinant CPG2 protein expression in BL21 *E. coli.* Coomassie Blue staining following SDS-PAGE of samples of whole cell lysate (lane 1), supernatant (lane 2) and re-suspended pellet (lane 3) of IPTG induced BL21 *E. coli* expressing CPG2 and CPG2-p700 proteins, respectively, from left to right, along with protein standards (ladder). The arrowheads indicate over-expressed CPG2 and CPG2-p700 proteins at the expected sizes (45 and 48 kDa, respectively) in the re-suspended pellets.

**Figure 2 molecules-26-00625-f002:**
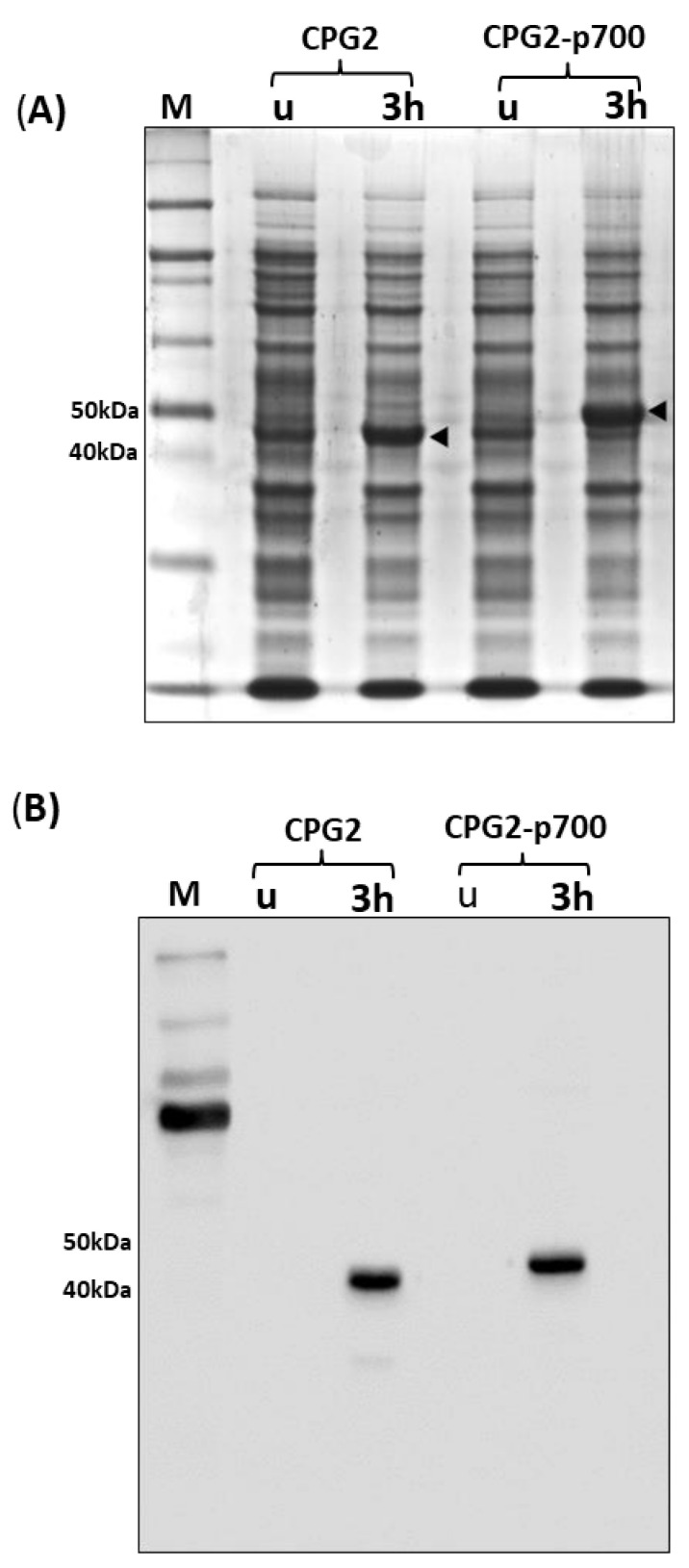
Large-scale expression of CPG2 recombinant proteins in BL21 *E. coli*. **Panel** (**A**) Coomassie blue staining following SDS-PAGE of paired samples (whole cell lysate) taken pre- (u) and 3 h post-IPTG induction (3 h) of CPG2, and CPG2-p700 recombinant protein expression, respectively, from left to right, compared with a ladder (extreme left). The arrowheads highlight the protein bands of the expected sizes (45 and 48 kDa). **Panel** (**B**) Western blotting analysis using antihistidine antibody on identical samples corresponding to the top panel, compared with a protein ladder (extreme left). Histidine-tagged recombinant proteins of the expected sizes (45 and 48 kDa) were detected in post-induction samples, but not in corresponding pre-induction samples.

**Figure 3 molecules-26-00625-f003:**
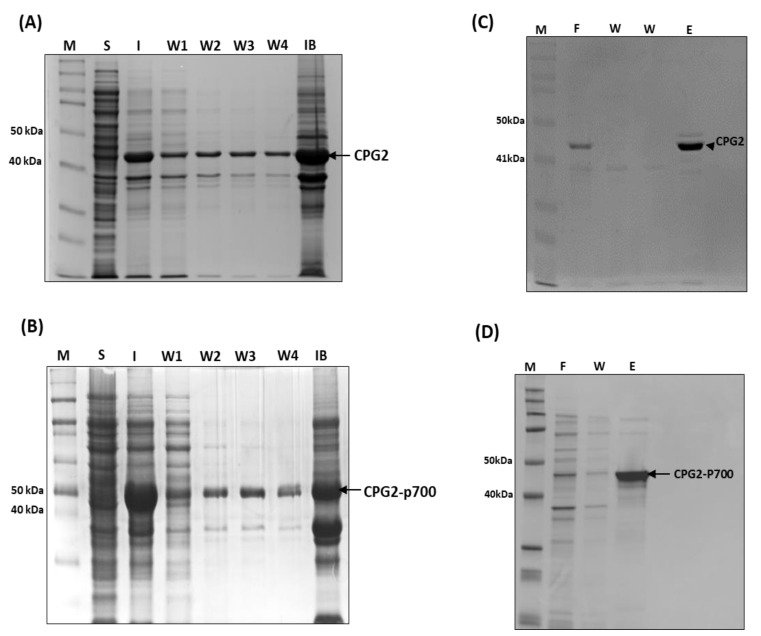
Purification of recombinant CPG2 proteins on nickel-chelate resin from urea washes derived from the insoluble fraction of BL21 *E. coli* cell lysates. (**A**) CPG2 and (**B**) CPG2-p700 coomassie blue stained SDS-PAGE gels. Lanes are of standard protein ladder; soluble fraction; insoluble fraction; four sequential urea washes; and remaining inclusion body, respectively, of IPTG-induced BL21 *E. coli* cells expressing. Pooled washes (1 to 4) from the insoluble fraction of lysates derived the IPTG-induced BL21 *E. coli* cells expressing His-tagged recombinant (**C**) CPG2 and (**D**) CPG2-p700 (right panel) were purified using a Ni2+ chelate column. Lanes (left to right) represent Coomassie Blue staining following SDS-PAGE of standard protein ladder, initial flow-through, wash and eluent (purified recombinant protein).

**Figure 4 molecules-26-00625-f004:**
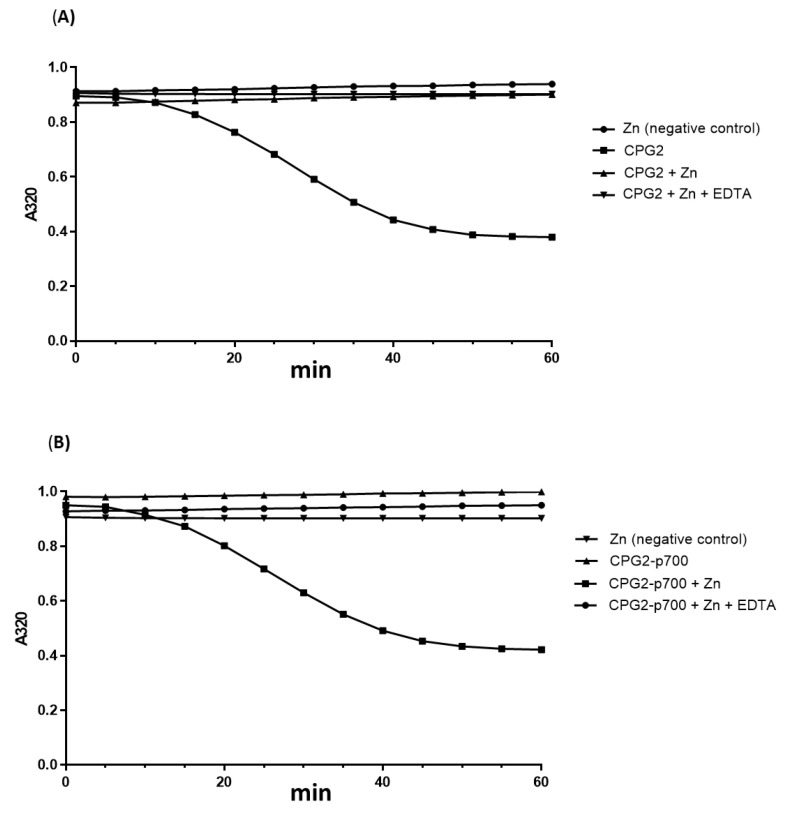
Confirmation of the zinc-dependent enzymatic activity of purified recombinant CPG2 proteins. Recombinant proteins were incubated with methotrexate at 37 °C for 0–60 min. The absorbance of each reaction at 320 nm was plotted against time. A similar reduction in absorbance (representing catalysis of MTX) was seen for both CPG2 (**A**) and CPG2-p700 (**B**) in the presence of ZnSO4 but not in its absence nor when the zinc chelating agent, EDTA, was added.

**Figure 5 molecules-26-00625-f005:**
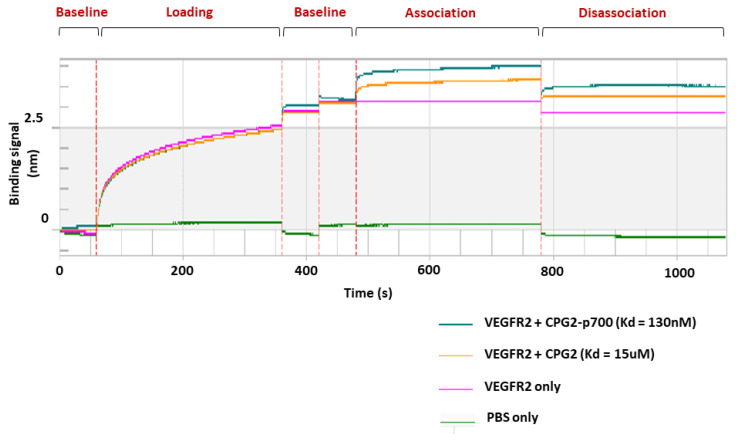
A representative biolayer interferometry experiment comparing the interaction between recombinant CPG2 or CPG2-p700 and VEGFR2. The binding signal is the wavelength shift detected by the BLItz machine and corresponds to the change in thickness of the biolayer as a result of the interaction between the recombinant proteins (CPG2 or CPG2-p700) and a VEGFR2-coated biosensor and is plotted against the duration of the interactions. Four runs are shown with PBS only loaded (green line), VEGFR2 only loaded (purple line) and VEGFR2 loaded followed by CPG2 (yellow line) or CPG2-p700 (green line). The first loading curve is the binding of VEGFR2-Fc to the protein A coated sensor. The sensor was then washed by dipping into PBS (baseline) before dipping into the CPG2 protein solutions (association). The sensor was then transferred to PBS and dissociation measured. Binding affinity (dissociation constant, Kd) was calculated using the BLItz software based on five repeats using increasing concentrations (300 nM to 10 mM) of recombinant proteins. CPG2-p700 showed a 100-fold increase in binding affinity compared with CPG2.

**Figure 6 molecules-26-00625-f006:**
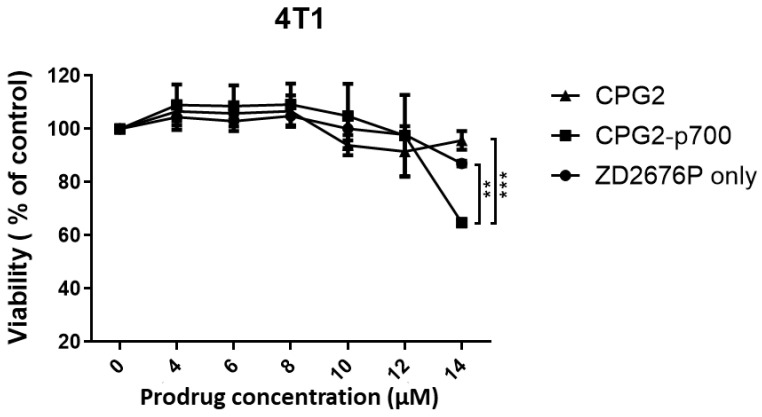
The effect of increasing concentrations of prodrug ZD2676P on the viability of the mouse breast cancer cell line 4T1 in the presence or absence of CPG2 or CPG2-p700. Data are means ± SEM; *n* = 3 (** = *p* < 0.01, *** = *p* < 0.001) indicates significance, two-way ANOVA, multiple comparison test.

## Data Availability

The data presented in this study are available in Appendix A.

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
