# Peer review of "Coupling of a Novel TIMP3 Peptide to Carboxypeptidase G2 for Pro-Drug Activation at the Tumour Site"

_molecules, 2021, doi:10.3390/molecules26030625_

Round 1

Reviewer 1 Report

This report by Mohammad et. Al. presented a p700-CPG2 fusion protein to target the tumor cell and activate potent nitrogen-mustard pro-drugs. Additionally, the authors demonstrated that p700-CPG2 had a 100-fold increase in binding affinity for VEGFR2 compared with CPG2 alone. In general, the results are significant, the manufacture of fusion protein are well designed. However, there are a few aspects that are worthy of further consideration:

1 p700 is a small peptide, so it is very likely to be degraded during the expression and purification, so SDS-PAGE is not sensitive enough to exclude the proteolysis of p700. ESI-MS or some sensitive methods should be used to check the mass of the fusion protein after the purification.      

2 The author demonstrated that p700 exhibits a broader binding specificity for growth factor receptors such as VEGFR1, VEGFR3, PDGFRα, FGFR1, FDFR2α, FDFR3 and FDFR4, but they just verified the binding between p700-COG2 to VEGFR2.

3 The in vitro test just included a MTT test in this work, which is far from enough to support the conclusion.

4 Which is the different between the Figure 6 and the cell viability test in Supplementary material.

Author Response

Dear Sir or Madam,

Title: Coupling of a novel TIMP3 peptide to carboxypeptidase G2 for pro-drug activation at the tumour site.

Thank you for the opportunity to respond to the referees’ comments regarding the above submitted manuscript. We will, however, attempt to address the referees’ comments as best we can in the response to reviewers’ comments below. These are shown in bold.

Reviewer 1:

1 p700 is a small peptide, so it is very likely to be degraded during the expression and purification, so SDS-PAGE is not sensitive enough to exclude the proteolysis of p700. ESI-MS or some sensitive methods should be used to check the mass of the fusion protein after the purification.     

After purification, we found the conjugate to still be intact which otherwise would have come out as different bands in the SDS page. Also, Biolayer interferometry showed that CPG2-p700 had a 100-fold increase in binding affinity for VEGFR2 compared with CPG2 alone which strongly suggest that the peptide wasn’t degraded after the purification. However, for future experiments we will defiantly consider the use of ESI-MS to evaluate the mass of the fusion protein post purification.

2 The author demonstrated that p700 exhibits a broader binding specificity for growth factor receptors such as VEGFR1, VEGFR3, PDGFRα, FGFR1, FDFR2α, FDFR3 and FDFR4, but they just verified the binding between p700-COG2 to VEGFR2.

As stated in the manuscript, a previous study from our lab has demonstrated the binding of p700 with the above mentioned receptor. However, we only aimed to test VEGFR2 binding of p700-CPG2 in this study as this is a major angiogenic receptor and we hope to subsequently test this in future studies.

3 The in vitro test just included a MTT test in this work, which is far from enough to support the conclusion.

The MTS test was carried out to evaluate the overall cytotoxic effect of the conjugate but not to actually probe the apoptotic mechanism.

4 Which is the different between the Figure 6 and the cell viability test in Supplementary material.

They are the same thing but the graphical abstract is included to give an overview of the whole study in the supplemental.

Reviewer 2 Report

In the manuscript entitled "Coupling of a novel TIMP3 peptide to carboxypeptidase G2 for pro-drug activation at the tumour site" the Authors report the use of antibody-directed enzyme prodrug therapy (ADEPT)-like molecule with the peptide sequence p700, derived from TIMP-3, fused to His-tagged, codon-optimized CPG2. The recombinant protein was amplified and analyzed in terms of binding affinity, catalytic activity and  cytotoxic activity in the 4T1 mouse breast cancer cell line. The manuscript is well written and the ADEPT is sufficiently characterized. Despite its chemical characterization is good, its impact in tumor biology must me improved.

Although the presented experiment is enough to validate the potential of the new  peptide, the characterization of the cytotoxic effect of the peptide+pro-drug must be improved. First, the impact in viability should be assayed in a human  breast cancer cell line. The pro-drug dose must be also increased, as the statistically significant differences are slight in all but the last dose (14uM).

Another issue that should be addressed is tu immunogenicity. Fusion proteins can lead to the formation of neo-antigens that lead to a strong immune response. Please use some of the public-available algorithms to check if the new sequence can render to a neo-antigen formation.

The analysis of this new therapeutic proposal in vivo could improve notably the novelty and scientific impact of this research. For future consideration, the biodistribution an toxicity evaluation of the new peptide and the xenograft tumor growth evaluation could add a lot of valuable information.

Author Response

Dear Sir or Madam:

Title: Coupling of a novel TIMP3 peptide to carboxypeptidase G2 for pro-drug activation at the tumour site.

Thank you for the opportunity to respond to the referees’ comments regarding the above submitted manuscript. We will, however, attempt to address the referees’ comments as best we can in the response to reviewers’ comments below. These are shown in bold.

Reviewer 2:

In the manuscript entitled "Coupling of a novel TIMP3 peptide to carboxypeptidase G2 for pro-drug activation at the tumour site" the Authors report the use of antibody-directed enzyme prodrug therapy (ADEPT)-like molecule with the peptide sequence p700, derived from TIMP-3, fused to His-tagged, codon-optimized CPG2. The recombinant protein was amplified and analyzed in terms of binding affinity, catalytic activity and cytotoxic activity in the 4T1 mouse breast cancer cell line. The manuscript is well written and the ADEPT is sufficiently characterized. Despite its chemical characterization is good, its impact in tumor biology must be improved.

Although the presented experiment is enough to validate the potential of the new peptide, the characterization of the cytotoxic effect of the peptide+pro-drug must be improved. First, the impact in viability should be assayed in a human breast cancer cell line. The pro-drug dose must be also increased, as the statistically significant differences are slight in all but the last dose (14uM).

This study is only a proof of concept and in future work we intend to increase concentration beyond what is published here and further test the apoptotic mechanism involved in a panel of cancer cell lines.

Another issue that should be addressed is tu immunogenicity. Fusion proteins can lead to the formation of neo-antigens that lead to a strong immune response. Please use some of the public-available algorithms to check if the new sequence can render to a neo-antigen formation.

As mentioned in the manuscript p700 is a human peptide making it unlikely to elicit immune response. Coupling a bacterial enzyme to such peptide will likely reduce any undesirable immune response as this is exemplified in different studies in the literature. However, we hope to test this in subsequent in vivo studies. Also, CPG2 has been used clinically over the years to reduce toxicity of methotrexate making it a safe approach in our study Trifilio et al. 2013. Clin Adv Hematol Oncol11(5), pp.322-323..

The analysis of this new therapeutic proposal in vivo could improve notably the novelty and scientific impact of this research. For future consideration, the biodistribution an toxicity evaluation of the new peptide and the xenograft tumor growth evaluation could add a lot of valuable information.

This is a valuable comment and we definitely are considering this in our future study.

Round 2

Reviewer 1 Report

I have no question any more.

Author Response

Thanks. 

Reviewer 2 Report

Dear authors,

The description and characterization of the new peptide is correct. Despite, in terms of its biological activity the differences are subtle. In this regard, an exploratory cell viability assay including more concentrated doses should be done, ideally including a human cell line, in order to elucidate if the differences are maintained using bigger doses. I consider this is a feasible experiment which would dramatically increase the experimental support for your approach.

Author Response

Thank you for the opportunity to respond to the referees’ comments regarding the above submitted manuscript. One of the referees (Reviewer 2) indicates that some further revisions are necessary in order to be accepted, including one new experiment (higher drug doses and testing on human cells).  While we would concur that this experiment would add weight to the conclusions, unfortunately it is not now possible as the principal investigator of the project has retired from research and the other authors are now working in different laboratories, without access to the original reagents.

If the editors fully agree with this decision, then we will be unable to resubmit this paper.  However, we would strongly argue that the requested experiment is not essential and the conclusions drawn stand without it. 

The paper seeks to determine whether a short peptide, shown previously to inhibit multiple proangiogenic growth factor receptors, can be used to specifically localise CPG2 to tumours, relative to unmodified CPG2, and whether this results in increased cytotoxicity and we have clearly shown this to be the case.

While testing the reagent at higher doses and on other tumour cells would be appropriate, we had originally intended that this would form part of a further study that included in vivo testing in mice. 4T1 cells were chosen here as we had previously shown p700 to be effective at inhibiting the growth of 4T1 tumours in a mouse breast cancer model. However, 4T1 cells are usually highly resistant to cytotoxic drugs so any killing is very significant. Moreover, the p700 sequence has been shown to bind to growth factor receptors of both species and so we have no reason to think the reagent would be less effective on human cells. Indeed, we have recently shown that p700 can effectively deliver liposomal doxorubicin to human breast cancer cells. We have now included this information and additional citations in the conclusion. https://www.mdpi.com/1420-3049/26/1/100

We cannot deny that the experiment proposed by this referee is worthwhile but, for the reasons highlighted above, we are unable to carry out and do not regard it as essential. We would greatly appreciate it, therefore, if you would accept the modified version of our paper in your respected journal without it.